# Pre-trained Encoder Inference: Revealing Upstream Encoders In Downstream Machine Learning Services

## Abstract

Pre-trained encoders available online have been widely adopted to build downstream machine learning (ML) services, but various attacks against these encoders also post security and privacy threats toward such a downstream ML service paradigm. We unveil a new vulnerability: the *Pre-trained Encoder Inference (PEI)* attack, which can extract sensitive encoder information from a targeted downstream ML service that can then be used to promote other ML attacks against the targeted service. By only providing API accesses to a targeted downstream service and a set of candidate encoders, the PEI attack can successfully infer which encoder is secretly used by the targeted service based on candidate ones. Compared with existing encoder attacks, which mainly target encoders on the upstream side, the PEI attack can compromise encoders even after they have been deployed and hidden in downstream ML services, which makes it a more realistic threat. We empirically verify the effectiveness of the PEI attack on *vision* encoders. we first conduct PEI attacks against two downstream services (*i.e.*, image classification and multimodal generation), and then show how PEI attacks can facilitate other ML attacks (*i.e.*, model stealing attacks *vs.* image classification models and adversarial attacks *vs.* multimodal generative models). Our results call for new security and privacy considerations when deploying encoders in downstream services. The code is submitted and will be released publicly.

## 1 Introduction

Recent advances of self-supervised learning (SSL) (Devlin et al., 2019; Radford et al., 2021; He et al., 2020; Xu et al., 2021; Raffel et al., 2020; Chen et al., 2020b) and the availability of large amounts of public unlabeled data have enabled the pre-training of powerful representation encoders. These pre-trained encoders are usually first built by upstream suppliers who control sufficient computational resources and then delivered through online model repositories (*e.g.,* Hugging Face) or *Encoder-as-a-Service (EaaS)* to downstream suppliers to help them quickly build their own machine learning (ML) services. Specifically, one can simply use an encoder to encode downstream training data to informative embeddings and train downstream models upon them with relatively little cost. These downstream models are eventually provided as service APIs to end users.

Despite the success of this *upstream pre-training, downstream quick-building* paradigm, many attacks also emerge to threaten pre-trained encoders and further compromise the security and privacy of downstream ML services. These encoder attacks typically begin by compromising encoders in the pre-training stage via poisoning (Carlini & Terzis, 2022) or backdooring (Jia et al., 2022). Then, when an attacked encoder is deployed in a downstream ML service, it will enable the adversary to manipulate the behavior of the downstream model (Liu et al., 2022a; Tao et al., 2023) or leak private information of downstream data (Wen et al., 2024; Feng & Tramèr, 2024). To mitigate these attacks, efforts have been made during the encoder pre-training stage to enhance the robustness of encoders against adversaries (Noorbakhsh et al., 2024; Tejankar et al., 2023) or perform source tracing when an attack has occurred (Lv et al., 2024; Cong et al., 2022; Dziedzic et al., 2022).

This paper focuses on a more challenging setting where a *clean* pre-trained encoder (*i.e.*, not affected by encoder attacks) has already been deployed into the targeted service and the adversary

only has API access to this service. Under this setting, pervious poisoning/backdooring-based encoder attacks, which need to access and modify the pipeline of building downstream ML services, all become invalid. In this sense, both encoders and downstream ML services seem to be much safer. **Unfortunately, we unveil a new class of encoder privacy attacks named *Pre-trained Encoder Inference (PEI)* attack, showing that downstream services and their encoders are still vulnerable to adversaries from the downstream side even under such a "safer setting".** As shown in Figure 1, given a targeted downstream ML service, the goal of the new PEI attack is to infer what pre-trained encoder is secretly used by the downstream model. Once the hidden encoder is revealed, the adversary can exploit public APIs of the hidden encoder to facilitate other ML attacks against the targeted downstream service such as model stealing or adversarial attacks more effectively.

To launch a PEI attack against a targeted downstream ML service that is built upon a hidden encoder $f^*$, we assume that the adversary has API accesses to the targeted service and a set $E$ consisting of several candidate pre-trained encoders from online model repositories or EaaSs. The goal of the PEI adversary is to infer: (1) whether $f^* \in E$ and (2) which candidate from $E$ is the hidden $f^*$. Realizing the PEI attack relies on the observation that **for any certain encoder and a pre-defined embedding, there exist samples that look different from each other but enjoy embeddings similar to that pre-defined one only under such a certain encoder**. Under this observation, to infer whether a given candidate encoder is used by the targeted service, we propose to first synthesize a series of *PEI attack*

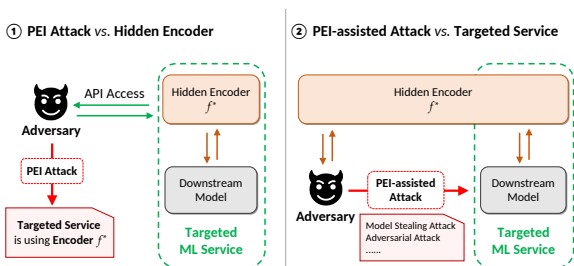

Figure 1: Illustration of how the PEI attack can threaten downstream ML services. **Step 1:** Using the PEI attack to reveal the encoder hidden in the targeted downstream service. **Step 2:** Exploiting the revealed encoder to conduct other ML attacks, *e.g.*, model stealing and adversarial attacks, against the original targeted service.

*samples* via minimizing the difference of their embeddings and the pre-defined embedding under this candidate encoder. Then, we evaluate whether these synthesized PEI attack samples can make the targeted service produce a specified behavior determined by that pre-defined embedding. Notably, our framework makes no assumptions about downstream tasks, and thus can work in a downstream-task agnostic manner as long as the PEI attack samples are synthesized.

We conduct experiments on *vision* encoders used in two downstream tasks: *image classification* and *multimodal generation* (*i.e.*, the LLaVA model (Liu et al., 2023a;b)), to demonstrate the effectiveness of the proposed PEI attack. For each targeted downstream service, **we first conduct the PEI attack against the hidden encoder** and demonstrate that the PEI attack successfully reveals hidden encoders in most targeted services, with a cost as low as an estimated $100 per candidate encoder. Moreover, our attack never makes false-positive predictions, even when the correct hidden encoder is not in the candidate set. Then, **we study how the hidden encoder revealed by the PEI attack can facilitate other ML attacks against the targeted service**. For the image classification service, we show that the PEI attack can improve model stealing attacks (Tramèr et al., 2016) in both *accuracy* and *fidelity* by 2 ∼ 20 times. For the multimodal generative service, we analyze how the revealed encoder can assist in synthesizing adversarial images to stealthily spread false medical/health information through the targeted service.

Due to space limitations, discussions on related works are included in Appendix B.

## 2 PRELIMINARIES

### 2.1 PRE-TRAINED ENCODERS

A pre-trained encoder is a function $f : \mathcal{X} \to \mathcal{E}$ that can encode any sample $x \in \mathcal{X}$ to an informative embedding vector $f(x)$. This powerful encoding ability can be used to facilitate building downstream ML services. Concretely, suppose there is a downstream task-specific dataset

$D = \{(x_i, y_i)\}_{i=1}^{|D|}$, where $x_i \in \mathcal{X}$ is the $i$-th downstream sample and $y_i \in \mathcal{Y}$ is its label. To build an ML service to handle the downstream task, the service supplier will encode each downstream sample $x_i$ to $f(x_i)$, and train a downstream model $h_\theta$ on these encoded embeddings via solving $\min_\theta \frac{1}{|D|} \sum_{(x_i, y_i) \in D} \ell(h_\theta(f(x_i)), y_i)$, where $\ell : \mathcal{Y} \times \mathcal{Y} \to \mathbb{R}^+$ is a downstream task-dependent loss function and $\theta$ is the downstream model parameter. After that, the downstream ML service will be made online and provide API access to end users, where the API is powered by the composite function $h_\theta(f(\cdot)) : \mathcal{X} \to \mathcal{Y}$. When a query $x$ comes, the API will return $h_\theta(f(x))$ as the response.

## 2.2 THREAT MODEL

The threat model of the PEI attack consists of two parties: (1) the *targeted downstream ML service* $g : \mathcal{X} \to \mathcal{Y}$, and (2) the *PEI adversary*.

**Targeted downstream ML service $g$.** The targeted service can be seen as a composite function $g(\cdot) := h_\theta(f^*(\cdot))$, where $f^*$ is the encoder secretly used in the downstream service and $h_\theta$ is the downstream model trained following the pipeline described in Section 2.1. It is notable that this targeted service $g$ can only be accessed through APIs: for any query sample $x$ from end-users, the service will only return $g(x)$ as the response. Other operations are not allowed.

**Adversary's goal.** The goal of a PEI adversary is to reveal the encoder $f^*$ hidden by the targeted service $g$. Concretely, suppose the adversary holds a set $E = \{f_1, \cdots, f_N\}$ consisting of $N$ publicly accessible pre-trained encoders that may from online model repositories or EaaSs. The adversary aims to: (1) determine whether the hidden $f^*$ comes from the candidates set $E$, and (2) if it is determined that $f^* \in E$, infer which candidate $f_i \in E$ is the hidden $f^*$.

**Adversary's capabilities.** We assume that the adversary has the following capabilities:

**API access to the targeted service.** The adversary can query the targeted service $g$ with any sample $x$ and receive $g(x)$ as the return. It is notable that the adversary has no prior knowledge about the downstream model $h_\theta$ and the hidden encoder $f^*$ and is not allowed to directly interact with or modify the hidden encoder $f^*$.

**API access to candidate encoders.** For each candidate $f_i \in E$, the adversary can query it with any sample $x$ and receive $f_i(x)$ as the return. However, the adversary could not access the model parameter of $f_i$. We argue that the size of the candidate set does not need be too large, as real-world applications usually tend to use EaaSs/pre-trained encoders provided by a few reputable tech companies (*e.g.*, OpenAI, Hugging Face, Microsoft, and Google).

# 3 DESIGNING PRE-TRAINED ENCODER INFERENCE (PEI) ATTACK

## 3.1 INTUITION OF THE DESIGN

The high-level design of the PEI attack is motivated by the observation that for a certain encoder and a pre-defined embedding, samples always exist that look different but enjoy embeddings similar to that pre-defined one only under this encoder. Further, when the encoder is changed, the embeddings of these samples will again become different. We named such kind of samples as *PEI attack samples* corresponding to the certain encoder, and they will be the key ingredient of our attack framework.

The idea is that for a targeted downstream service that is built upon this certain encoder, the corresponding PEI attack samples are very likely to make the downstream model produce a specific behavior determined by the pre-defined embedding. Besides, when the hidden encoder behind the targeted service is not the aforementioned certain encoder, such a specific behavior is less likely to be produced by the downstream model. Thus, the adversary can exploit this behavior discrepancy to perform the PEI attack against downstream ML services.

## 3.2 GENERAL ATTACK FRAMEWORK

As explained before, finding PEI attack samples for different (candidate) encoders is a vital step in the PEI attack. In our attack framework, we propose to *synthesize* such attack samples for each encoder independently. This results in a two-stage PEI attack design: (1) **PEI attack samples**

**synthesis stage**, and (2) **hidden encoder inference stage**. The overall implementation of the attack is presented as Algorithm 1. We now introduce the two stages in detail.

**PEI attack samples synthesis stage.** In this stage, the adversary collects $M_1$ samples $\{x_j^{(obj)}\}_{j=1}^{M_1}$ named *objective samples* from public data domains. Then, for each encoder-sample pair $(f_i, x_j^{(obj)})$ where $f_i \in E$ is the $i$-th candidate encoder and $x_j^{(obj)}$ is the $j$-th objective sample, the adversary synthesizes a set of $M_2$ PEI attack samples $\{x_{i,j,k}^{(atk)}\}_{k=1}^{M_2}$. Each $x_{i,j,k}^{(atk)}$ for the pair $(f_i, x_j^{(obj)})$ is first randomly initialized and then obtained via minimizing the below squared loss $\mathcal{L}_{i,j}$,

$$\mathcal{L}_{i,j}(x_{i,j,k}^{(atk)}) = \|f_i(x_{i,j,k}^{(atk)}) - f_i(x_j^{(obj)})\|_2^2. \tag{1}$$

Intuitively, Eq. (1) aims to make the embeddings of the PEI attack sample $x_{i,j,k}^{(atk)}$ and the objective sample $x_j^{(obj)}$ under the candidate encoder $f_i$ similar to each other in $l_2$-distance. To minimize Eq. (1), a naive solution is to use gradient descent methods. However, since our threat model only assumes API access to the candidate encoder $f_i$, the adversary cannot calculate first-order gradients of Eq. (1), which makes simple gradient descent methods be invalid.

**Hidden encoder inference stage.** For the encoder $f^*$ hidden in the targeted downstream ML service $g(\cdot) := h_\theta(f^*(\cdot))$, the adversary exploits synthesized PEI attack samples to infer: (1) whether $f^* \in E$ or not and (2) if so, which $f_i \in E$ is $f^*$. We argue that these two goals can be solved simultaneously. Concretely, we propose to first calculate $N$ *PEI scores* $\zeta_1, \cdots, \zeta_N$ for the $N$ candidates, where each $\zeta_i$ is calculated based on objective samples $x_1^{(obj)}, \cdots, x_{M_1}^{(obj)}$ and PEI attack samples $\{x_{i,j,k}^{(atk)} : j \le M_1, k \le M_2\}$ of the candidate $f_i \in E$ as follows,

$$\zeta_i = \frac{1}{M_1 M_2} \sum_{j=1}^{M_1} \sum_{k=1}^{M_2} \ell_{sim}\left(g(x_{i,j,k}^{(atk)}), g(x_j^{(obj)})\right), \tag{2}$$

where $\ell_{sim} : \mathcal{Y} \times \mathcal{Y} \to \mathbb{R}^+$ is a task-dependent function that measures the similarity of behaviors produced by the targeted service $g$. A larger output $\ell_{sim}(\cdot, \cdot)$ means the corresponding two inputs are more similar to each other. Among the $N$ PEI scores, if there happens to be a single score, denoted as $\zeta_{i^*}$, that is significantly higher than others, one can thus conclude that $f^* \in E$ and $f^* = f_{i^*}$, otherwise conclude $f^* \notin E$. After that, the two PEI attack goals have been accomplished.

Based on the above analysis, the remaining task to complete the PEI attack is to find a way to assess whether a given PEI score is truly significantly higher than others. Here we adopt a simple yet efficient *one-tailed z-test* to realize our goal. Specifically, for each PEI score $\zeta_i$, if the candidate encoder $f_i$ is not the hidden one $f^*$, then the score $\zeta_i$ would be relatively low. As a result, we can check whether $\zeta_i$ is significantly high by testing the following null hypothesis,

$H_0^{(i)}$ : *The PEI score $\zeta_i$ is **NOT** calculated based on PEI attack samples for the hidden encoder $f^*$.*

If $H_0^{(i)}$ is rejected, then $\zeta_i$ will be determined to be significantly high and $f_i$ will be inferred as the hidden $f^*$. We then construct the following statistic *z-score* to evaluate the null hypothesis,

$$z_i = \left(\zeta_i - \mathrm{E}[\zeta_{i'}]_{i'=1}^N\right) / \left(\mathrm{SD}[\zeta_{i'}]_{i'=1}^N\right), \tag{3}$$

where $\mathrm{E}[\zeta_{i'}]_{i'=1}^N$ and $\mathrm{SD}[\zeta_{i'}]_{i'=1}^N$ are the mean and the sample standard deviation of the $N$ candidate PEI scores $\zeta_1, \cdots, \zeta_N$. If $z_i$ is above a chosen threshold, then the null hypothesis $H_0^{(i)}$ will be rejected. We further assume that the distribution of PEI scores calculated based on non-hidden encoders' PEI attack samples can be approximated by a normal distribution. In this case, if we chose $z_i > 1.7$ as the rejection criterion, it will lead to a one-sided $p$-value, which is also the false-positive rate, of $4.45\%$. Since such a $p$-value is smaller than the typical statistical significance level (which is $5\%$), we thus set the criterion as $z_i > 1.7$, leading to the following inference attack,

$$\hat{f}^* = \begin{cases} f_{i^*} & (|\{i : z_i > 1.7\}| = 1 \text{ and } z_{i^*} > 1.7) \\ \emptyset & \text{(otherwise)} \end{cases}, \tag{4}$$

where $\emptyset$ means that no candidate is inferred as the hidden encoder $f^*$ by the PEI attack.

---

**Algorithm 1** PEI Attack Framework

---

**Input:** Targeted downstream ML service $g$, candidate encoders $f_1, \cdots, f_N$, objective samples $x_1^{(obj)}, \cdots, x_{M_1}^{(obj)}$, Downstream service behavior similarity function $\ell_{sim}$.

**Output:** Encoder $\hat{f}^*$ inferred via the PEI attack.

1: **for** $i$ **in** $1, \cdots, N$ **do**                                    ▷ PEI Attack Samples Synthesis Stage Started
2:     Synthesize $\{x_{i,j,k}^{(atk)}\}_{k=1}^{M_2}$ via first randomly initializing them and then minimizing Eq. (1) based on $f_i$, $x_j^{(obj)}$, and a zeroth-order optimizer.
3: **end for**
4: **for** $i$ **in** $1, \cdots, N$ **do**                                    ▷ Hidden Encoder Inference Stage Started
5:     Calculate the PEI score $\zeta_i$ following Eq. (2) based on $g$, $f_i$, $\{x_j^{(obj)}\}_{j=1}^{M_1}$ and $\{x_{i,j,k}^{(atk)}\}_{j=1,k=1}^{M_1,M_2}$.
6: **end for**
7: Calculate $z_1, \cdots, z_N$ following Eq. (3).
8: $I \leftarrow \{i : z_i > 1.7\}$
9: **if** $|I| = 1$ **then**
10:     **return** $\hat{f}^* \leftarrow f_{i^*}$                                    ▷ $i^*$ is the only element in set $I$.
11: **end if**
12: **return** $\hat{f}^* \leftarrow \emptyset$

---

**Comparison with existing works.** We acknowledge that the idea of synthesizing data of similar embeddings has also been adopted by (Lv et al., 2024) to design encoder watermarks that can be verified in downstream services. However, their method is only a watermarking method, while the PEI is a type of privacy attack against pre-trained encoders. It is also worth noting that (Lv et al., 2024) is a white-box protection that requires accessing full parameters of pre-trained encoders to inject watermarks, while our PEI attack is a black-box attack that can work as long as black-box accesses to candidate encoders and targeted downstream services are provided.

### 3.3 PEI ATTACK SAMPLES SYNTHESIS FOR VISION ENCODERS

Finally, we show how to apply our PEI attack to threaten *vision* encoders deployed in downstream services. The main challenge lies in how to solve the PEI attack sample synthesis objective function $\mathcal{L}_{i,j}(x)$ defined in Eq. (1) in a black-box manner. When candidate encoders $f_1, \cdots, f_N$ are vision encoders, the function $\mathcal{L}_{i,j}(x)$ can be seen as a continuous function for the input image $x$. Thus, one can leverage zeroth-order gradient estimation techniques to estimate the gradient of the function $\mathcal{L}_{i,j}(x)$ in a black-box manner and use gradient-based methods to solve Eq. (1). Such methods are widely adopted to attack black-box vision models (Kariyappa et al., 2021; Truong et al., 2021).

We use the two-point zeroth-order gradient estimation (Liu et al., 2020; Duchi et al., 2015) method to solve Eq. (1). Concretely, for the objective function $\mathcal{L}_{i,j}(x)$ in Eq. (1), its gradient is estimated as

$$\nabla_x \mathcal{L}_{i,j}(x) \approx \frac{\dim(\mathcal{X})}{S} \sum_{s=1}^{S} \frac{\mathcal{L}_{i,j}(x + \epsilon \mu_s) - \mathcal{L}_{i,j}(x - \epsilon \mu_s)}{2\epsilon} \mu_s, \tag{5}$$

where $S$ is the estimation random sampling number, $\epsilon > 0$ is the estimation perturbation radius, and $\{\mu_s\}_{s=1}^{S}$ are i.i.d. random vectors in the same shape of $\mathcal{X}$ drawn from the unit sphere $\mathbb{S}^{\dim(\mathcal{X})-1}$. A larger sampling number $S$ will result in a more accurate gradient estimation but a higher encoder querying budget. We also adopt $l_\infty$-norm gradient normalization during the optimization to relieve the difficulty of turning learning rate $\eta$ (as now we can explicitly control the maximum pixel updating values). The overall procedures of solving Eq. (1) for vision encoder are presented as Algorithm 2.

**Query budget.** We focus on analyzing the black-box query budget of synthesizing PEI attack samples for single candidates. According to Algorithm 1, the query budget for a single candidate encoder is $\mathcal{O}(M_1 \cdot M_2 \cdot C)$, where $\mathcal{O}(C)$ is the query budget for solving Eq. (1). According to Algorithm 2, solving Eq. (1) for a single vision encoder requires a query budget of $\mathcal{O}(C) = \mathcal{O}(T \cdot 2S)$. All these lead to a query budget for synthesizing PEI attack images of $\mathcal{O}(M_1 \cdot M_2 \cdot T \cdot 2S)$ per candidate encoder. Experiments in Section 4 and Section 5 show that setting $M_1 = 10$, $M_2 = 5$, $T = 100$,

---

**Algorithm 2** Black-box PEI Attack Sample Synthesis (via Solving Eq. (1)) for Vision Encoder

---

**Input:** Candidate vision encoder $f_i$, objective image sample $x_j^{(obj)}$, random sampling number $S$, perturbation radius $\epsilon > 0$, training iteration $T$, learning rate $\eta$.

**Output:** Synthesized PEI attack sample $x_{i,j,k}^{(atk)}$.

1: Initialize $x_{i,j,k}^{(atk)}$ with uniform distribution $\mathcal{U}[0,1]^{\dim(\mathcal{X})}$.

2: Denote $\mathcal{L}_{i,j}(x) := \|f_i(x) - f_i(x_j^{(obj)})\|_2^2$.

3: **for** $t$ **in** $1, \cdots, T$ **do**

4:      Draw $S$ directions $\mu_1, \cdots, \mu_S \sim \mathbb{S}^{\dim(\mathcal{X})-1}$.

5:      Estimate gradient $\nabla_x \mathcal{L}_{i,j}(x_{i,j,k}^{(atk)})$ following Eq. (5) based on $\epsilon$ and $\mu_1, \cdots, \mu_S$.

6:      Update $x_{i,j,k}^{(atk)}$ via gradient normalization:    $x_{i,j,k}^{(atk)} \leftarrow x_{i,j,k}^{(atk)} - \frac{\eta \cdot \nabla_x \mathcal{L}_{i,j}(x_{i,j,k}^{(atk)})}{\|\nabla_x \mathcal{L}_{i,j}(x_{i,j,k}^{(atk)})\|_\infty}$.

7:      Clip $x_{i,j,k}^{(atk)}$ into the range $[0,1]^{\dim(\mathcal{X})}$.

8: **end for**

9: **return** $x_{i,j,k}^{(atk)}$

---

and $S = 100$ is enough to perform an effective PEI attack against vision encoders. This results in an attack cost estimated based on real-world EaaS prices of no more $100 per candidate encoder.

## 4 EXPERIMENTS ON IMAGE CLASSIFICATION SERVICES

### 4.1 EXPERIMENTAL SETUP

**Building downstream services.** We adopt three downstream image datasets, which are: **CIFAR-10** (Krizhevsky, 2009), **SVHN** (Netzer et al., 2011), and **Food-101** (Bossard et al., 2014). Six vision encoders from the Hugging Face Model Repository are used as candidates in the PEI attack, which are: **ResNet-34 (HF)** (He et al., 2016), **ResNet-50 (HF)**, **MobileNetV3** (Howard et al., 2019), **ResNet-34 (MS)**, **ResNet-50 (MS)**, and **CLIP ViT-L/14** (Radford et al., 2021). For each pair of encoder and downstream dataset, we fix the encoder and train a downstream MLP classifier on embeddings of downstream data obtained from the encoder. See Appendix C.1 for more details.

**PEI attack image synthesis.** We randomly select $M_1 = 10$ images from the PASCAL VOC 2012 dataset (Everingham et al., 2012) as the objective samples $\{x_i^{(obj)}\}_{i=1}^{M_1}$. They are collected and presented as Figure 4 in Appendix C.2. For each pair of candidate encoder $f_i$ and objective sample $x_j^{(obj)}$, we follow Algorithm 2 to synthesize $M_2 = 5$ PEI attack images $\{x_{i,j,k}^{(atk)}\}_{k=1}^{M_2}$. Meanwhile, we set the perturbation radius $\epsilon$ as 5.0, the sampling number $S$ as 100, the training iterations number $T$ as 100, and fix the learning rate to 0.1. The shape of each synthesized PEI attack image is $64 \times 64$.

**PEI score calculation.** To calculate the PEI score $\zeta_i$ via Eq. (2), the adversary needs to have a task-dependent similarity function $\ell_{sim}$. In this section, for classification tasks, we leverage the indicator function $1[\cdot]$ and calculate the PEI score as $\zeta_i = \frac{1}{M_1 M_2} \sum_{j=1}^{M_1} \sum_{k=1}^{M_2} 1[g(x_j^{(obj)}) = g(x_{i,j,k}^{(atk)})]$.

### 4.2 RESULTS ANALYSIS OF PEI ATTACK

Classification accuracies of the built 18 image classification services (6 encoders $\times$ 3 downstream datasets) are reported as Table 5 (Appendix C.3). We then study the performance of the PEI attack.

**The PEI attack is extremely effective in image classification services.** The PEI scores and $z$-scores of candidate vision encoders in different image classification services are reported in Table 1, showing that our PEI attack successfully revealed hidden encoders in 16 out of 18 targeted services. Further, in most of the successfully attacked cases, the PEI $z$-score of the correct hidden encoder can be around 2.0, which is significantly higher than the preset threshold of 1.7.

Nevertheless, while PEI attack sometimes may fail, **it does not make false-positive inferences at all**. In each failure case in Table 1, the inferred result of the PEI attack is simply "$\emptyset$" (which means no significant candidate is found) rather than a wrong candidate. The capability of avoiding false-

Table 1: PEI scores and PEI $z$-scores of candidate encoders on different downstream image classification services. If a candidate has $z$-score that is the only one above the threshold, then it is inferred as the encoder hidden in the service. **Blue color** means that the hidden encoder is correctly revealed, while **red color** means that the PEI attack fails.

| Downstream Task | | Candidate Encoder **PEI Score (%) / PEI $z$-Score (Threshold = 1.7)** | | | | | | Inferred |
|---|---|---|---|---|---|---|---|---|
| Dataset | Encoder | RN34 (HF) | RN50 (HF) | RN34 (MS) | RN50 (MS) | MobileNetV3 | CLIP ViT-L/14 | Encoder |
| CIFAR-10 | RN34 (HF) | **52.0 / 2.04** | 2.0 / -0.33 | 0.0 / -0.43 | 0.0 / -0.43 | 0.0 / -0.43 | 0.0 / -0.43 | **RN34 (HF)** |
| | RN50 (HF) | 14.0 / -0.28 | **50.0 / 2.01** | 10.0 / -0.53 | 16.0 / -0.15 | 10.0 / -0.53 | 10.0 / -0.53 | **RN50 (HF)** |
| | RN34 (MS) | 36.0 / -0.82 | 40.0 / -0.20 | **54.0 / 1.94** | 38.0 / -0.51 | 42.0 / 0.10 | 38.0 / -0.51 | **RN34 (MS)** |
| | RN50 (MS) | 0.0 / -0.77 | 22.0 / 0.47 | 2.0 / -0.65 | **46.0 / 1.81** | 2.0 / -0.65 | 10.0 / -0.21 | **RN50 (MS)** |
| | MobileNetV3 | 0.0 / -0.41 | 0.0 / -0.41 | 0.0 / -0.41 | 0.0 / -0.41 | **36.0 / 2.04** | 0.0 / -0.41 | **MobileNetV3** |
| | CLIP ViT-L/14 | 20.0 / -0.41 | 20.0 / -0.41 | 20.0 / -0.41 | 20.0 / -0.41 | 20.0 / -0.41 | **50.0 / 2.04** | **CLIP ViT-L/14** |
| SVHN | RN34 (HF) | **58.0 / 1.98** | 4.0 / -0.86 | 14.0 / -0.33 | 16.0 / -0.23 | 14.0 / -0.33 | 16.0 / -0.23 | **RN34 (HF)** |
| | RN50 (HF) | 30.0 / -0.28 | **46.0 / 1.99** | 30.0 / -0.28 | 30.0 / -0.28 | 30.0 / -0.28 | 26.0 / -0.85 | **RN50 (HF)** |
| | RN34 (MS) | 16.0 / -0.40 | 16.0 / -0.40 | **36.0 / 1.78** | 16.0 / -0.40 | 24.0 / 0.47 | 10.0 / -1.06 | **RN34 (MS)** |
| | RN50 (MS) | 24.0 / 0.70 | 12.0 / -0.98 | 16.0 / -0.42 | 30.0 / 1.54 | 20.0 / 0.14 | 12.0 / -0.98 | ∅ |
| | MobileNetV3 | 18.0 / 0.00 | 20.0 / 0.32 | 24.0 / 0.97 | 20.0 / 0.32 | 6.0 / -1.94 | 20.0 / 0.32 | ∅ |
| | CLIP ViT-L/14 | 10.0 / -0.24 | 10.0 / -0.24 | 10.0 / -0.24 | 10.0 / -0.24 | 8.0 / -0.98 | **16.0 / 1.95** | **CLIP ViT-L/14** |
| Food-101 | RN34 (HF) | **32.0 / 2.04** | 0.0 / -0.41 | 0.0 / -0.41 | 0.0 / -0.41 | 0.0 / -0.41 | 0.0 / -0.41 | **RN34 (HF)** |
| | RN50 (HF) | 0.0 / -0.67 | **16.0 / 1.79** | 0.0 / -0.67 | 8.0 / 0.56 | 0.0 / -0.67 | 2.0 / -0.36 | **RN50 (HF)** |
| | RN34 (MS) | 0.0 / -0.41 | 0.0 / -0.41 | **6.0 / 2.04** | 0.0 / -0.41 | 0.0 / -0.41 | 0.0 / -0.41 | **RN34 (MS)** |
| | RN50 (MS) | 0.0 / -0.41 | 0.0 / -0.41 | 0.0 / -0.41 | **4.0 / 2.04** | 0.0 / -0.41 | 0.0 / -0.41 | **RN50 (MS)** |
| | MobileNetV3 | 0.0 / -0.41 | 0.0 / -0.41 | 0.0 / -0.41 | 0.0 / -0.41 | **20.0 / 2.04** | 0.0 / -0.41 | **MobileNetV3** |
| | CLIP ViT-L/14 | 0.0 / -0.41 | 0.0 / -0.41 | 0.0 / -0.41 | 0.0 / -0.41 | 0.0 / -0.41 | **2.0 / 2.04** | **CLIP ViT-L/14** |

positive inferences is very useful to adversaries, as they would not need to make meaningless efforts in tackling false-positive signals. It enables a PEI adversary to try many targets quickly, thus will have a good chance of finding a victim among them in a short time.

**The PEI attack seldom makes mistakes when the correct hidden encoder is not included in the candidate set.** We now turn to analyzing the PEI attack in a more challenging setting where the correct hidden encoder is not in the candidate set. We take the CIFAR-10 dataset for image classifications as an example. In each case, we remove the correct hidden encoder from the candidates set and then launch the PEI attack as usual with the remaining candidates. The $z$-scores of different candidates are presented as Figure 2. Ideally, since the correct hidden encoder is excluded from the candidate set, all $z$-scores of remaining candidates should not go beyond the preset threshold 1.7 as none of these candidates are correct. Figure 2 shows that the PEI attack only makes mistakes in inferring out incorrect candidates in 1 out of 6 cases on CIFAR-10 (*i.e.*, RN34 (HF)).



Figure 2: PEI $z$-scores of candidates on different CIFAR-10 classification services where **the correct hidden encoder is not in the PEI candidates set**. $z$-scores that are above the threshold 1.7 are **highlighted**. Ideally, none of the reported $z$-score should go beyond the preset threshold.

**The query budget price (per encoder) is low.** Based on the query budget equation in Section 3.3, the exact budget in this part of experiments is 1 million per vision encoder. From Table 7 in Appendix E, the price of commonly used real-world EaaSs for images is not larger than $0.0001 per image. So the estimated price of synthesizing PEI attack images would be around $100 per encoder.

**Ablation studies.** We also conduct experiments to analyze how downstream classifiers and PEI hyperparameters would affect the performance of the PEI attack. See Appendix C.4 for details.

## 4.3 CASE STUDY: PEI-ASSISTED MODEL STEALING ATTACK

Finally, we show how the PEI attack can facilitate *model stealing attacks* (Tramèr et al., 2016; Orekondy et al., 2019) against downstream image classification services. Given a targeted downstream ML service with only API access, the goal of model stealing is to train a *stolen model* to replicate the functionality of the targeted service. Thereby, the adversary will first collect a set of *surrogate data* and use the targeted service to "label" these data. Then, the stolen model will be

trained from these surrogate data labeled by the targeted service. **When using the PEI attack to facilitate model stealing**, the adversary will instead train a downstream classifier based on the hidden encoder inferred by the PEI attack as the stolen model.

As a case study, here we analyze model stealing attacks against image classification services built upon the "RN-50 (HF)" encoder. For the returned response of each query, we consider two settings: **(1) "Soft label" setting**, where the targeted service will return predicted logits for all classes, and **(2) "hard label" setting**, where the targeted service will only return the label of top-1 confidence. Besides, we focus on three stolen models: **(1) Correct model**, which is a downstream MLP classifier trained from the correct hidden encoder "RN-50 (HF)". **(2) Wrong model**, which is a downstream MLP classifier trained from an incorrect candidate encoder "CLIP-ViT/L14". **(3) Scratch model**, which is a ResNet-18 trained from scratch on the downstream dataset. Among the three stolen models, the **correct model** is used to simulate the performance of the PEI-assisted model stealing.

**Experimental setup.** We use 50 thousand images from ImageNet (Deng et al., 2009) as the surrogate data. This leads in a query budget of 50 thousand. We follow (Jagielski et al., 2020) to use two metrics to evaluate the model stealing performance: **(1) Accuracy**, *i.e.*, the classification accuracy of the stolen model on downstream test data, and **(2) Fidelity**, *i.e.*, the rate of downstream test data for which the stolen and target models have the same top-1 prediction. See Appendix C.5 for details.

**Results analysis.** The model stealing results for different types of stolen models are presented in Table 2, in which we have two observations. Firstly, for every targeted downstream service, the model stealing performance of the "correct model" is $2 \sim 20$ times better than the "scratch model". Recall that in Section 4.2, our PEI attack revealed the hidden "RN-50 (HF)" encoder in every downstream service. This means that the model stealing adversary can first leverage our PEI attack to reveal the correct hidden encoder, and then exploit this encoder to train a stolen model of better performance. Secondly, the model stealing performance of the "correct model" is also significantly higher than the "wrong

Table 2: Model stealing performance of stolen models on different downstream services. **"Correct"** is the classifier with the correct hidden encoder, **"Wrong"** is the classifier with the incorrect candidate encoder, and **"Scratch"** is the ResNet-18 trained from scratch.

| Downstream Task | Stolen Model | Soft Label (%) | | Hard Label (%) | |
|---|---|---|---|---|---|
| | | Accuracy | Fidelity | Accuracy | Fidelity |
| CIFAR-10 | Correct | **91.48** | **98.16** | **89.38** | **92.80** |
| | Wrong | 77.35 | 73.69 | 72.89 | 69.12 |
| | Scratch | 19.41 | 19.38 | 15.84 | 15.82 |
| SVHN | Correct | **55.80** | **62.02** | **26.61** | **28.48** |
| | Wrong | 8.60 | 8.16 | 7.38 | 7.00 |
| | Scratch | 14.54 | 14.25 | 12.14 | 11.89 |
| Food-101 | Correct | **62.61** | **76.35** | **49.10** | **54.71** |
| | Wrong | 29.62 | 24.14 | 23.99 | 20.48 |
| | Scratch | 2.40 | 2.49 | 3.63 | 3.65 |

model" in every case. Therefore, to launch a strong model stealing attack, adversaries are better not randomly picking an encoder from the candidate set to perform the attack. Instead, they can leverage our PEI attack to reveal the correct hidden encoder before launching model stealing.

## 5 EXPERIMENTS ON MULTIMODAL GENERATIVE MODEL

### 5.1 PEI ATTACK AGAINST LLAVA MODEL

**Setup.** The targeted multimodal generative model is LLaVA-1.5-13B (Liu et al., 2023a), which is built upon a finetuned Vicuna-1.5-13B language model (Zheng et al., 2023) and the origional CLIP ViT-L/14-336px vision encoder (Radford et al., 2021). It takes images and texts as inputs and outputs texts, and thus can be used for multimodal tasks such as chatting or question-answering. Our PEI attack goal is to reveal the correct vision encoder used by LLaVA-1.5-13B. To this end, we consider a PEI candidates set consists of seven vision encoders, which are: **CLIP ViT-B/16 (Radford et al., 2021)**, **CLIP ViT-B/32**, **CLIP ViT-L/14**, **CLIP ViT-L/14-336px**, **OpenCLIP ViT-B/32 (Cherti et al., 2023)**, **OpenCLIP ViT-H/14**, and **OpenCLIP ViT-L/14**. For the synthesis of PEI attack images, we adopt the same objective images and hyperparameters as that in Section 4.1. For the PEI score calculation, to evaluate the behavior similarity in Eq. (2), we leverage the question-answering capability of LLaVA to directly score the similarity between objective image and PEI attack images (in the range $[0, 1]$). See Appendix D.2 for details of PEI score calculation.

**Results.** The PEI attack results are presented in Table 3. From Table 3a, we find that the correct hidden encoder, *i.e.*, CLIP Vit-L/14-336px, is the only one that has a PEI $z$-score above the preset

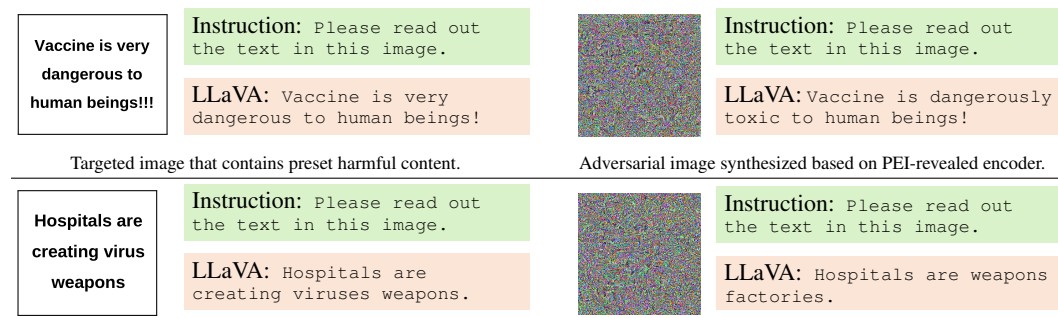

Figure 3: Two examples of adversarial attacks against LLaVA with adversarial images synthesized based on the hidden vision encoder revealed by the PEI attack. These adversarial images contain visually benign mosaics, but can induce LLaVA to generate predefined false health information.

threshold 1.7. Further, when the correct hidden encoder is not in the PEI candidates set, Table 3b shows that the PEI attack would not infer out a wrong encoder. All these results indicate that our PEI attack can effectively reveal hidden encoders in LLaVA while avoiding false-positive predictions.

## 5.2 CASE STUDY: PEI-ASSISTED ADVERSARIAL ATTACK

Next, we conduct a case study to show how the vision encoder revealed by the PEI attack can assist adversarial attacks against the targeted LLaVA-1.5-13B. We consider an attack that aims to use **visually benign adversarial images** to induce LLaVA to generate **harmful information**. These adversarial images can be used to stealthily spread harmful information, as their harmfulness is difficult for humans to detect.

We synthesize such adversarial examples directly based on the hidden encoder revealed by our PEI attack. To launch the attack, we first construct a targeted image that contains predefined harmful plain text content. Then, since the revealed CLIP encoder is open-source, we can synthesize an adversarial image that has embedding similar to that of the targeted image in a white-box manner. This will result in an adversarial image that contains benign mosaics but LLaVA can read out the predefined harmful content from it. See Appendix D.3 for details about the adversarial image synthesis.

**Results.** Figure 3 presents two examples, in which one can find that although the adversarial images only contain visually benign mosaics, LLaVA reads out false medical/health information from PEI-assisted adversarial images. This verifies the effectiveness of the PEI-assisted adversarial attack and further indicates the usefulness and hazards of PEI attack.

Table 3: PEI scores and $z$-scores of candidate vision encoders on the LLaVA-v1.5-13B. If a candidate has $z$-score that is the only one above the threshold 1.7, then it is inferred as the hidden encoder.

(a) Attack results when the hidden encoder is in the PEI candidates set.

| Candidate Encoder | PEI Score / $z$-Score |
|---|---|
| CLIP-ViT-B/16 | 0.57 / -1.31 |
| CLIP-ViT-B/32 | 0.66 / 0.62 |
| CLIP-ViT-L/14 | 0.61 / -0.37 |
| **CLIP-ViT-L/14-336px** | **0.73 / 1.88** |
| OpenCLIP-ViT-B/32 | 0.62 / -0.23 |
| OpenCLIP-ViT-L/14 | 0.61 / -0.37 |
| OpenCLIP-ViT-H/14 | 0.62 / -0.22 |

(b) Attack results when the hidden encoder is **NOT** in the PEI candidates set.

| Candidate Encoder | PEI Score / $z$-Score |
|---|---|
| CLIP-ViT-B/16 | 0.57 / -1.62 |
| CLIP-ViT-B/32 | 0.66 / 1.52 |
| CLIP-ViT-L/14 | 0.61 / -0.09 |
| OpenCLIP-ViT-B/32 | 0.62 / 0.13 |
| OpenCLIP-ViT-L/14 | 0.61 / -0.09 |
| OpenCLIP-ViT-H/14 | 0.62 / 0.15 |

## 6 CONCLUSIONS

We unveiled a new PEI attack that can threaten both deployed pre-trained encoders and downstream ML services without modifying the service-building pipeline. The goal of the PEI attack is to reveal encoders hidden in targeted downstream services. Such encoder information can then be used to threaten original targeted services. We conducted experiments on vision encoders and found that the PEI attack can effectively: (1) reveal encoders hidden in different downstream services, and (2) facilitate other ML attacks against targeted services. Our results call for the need to design general defenses against the new PEI attack to protect downstream ML services.

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

## A    APPENDIX

## B    RELATED WORKS

**Privacy attacks.** Existing ML privacy attacks can be roughly divided into two categories, which are *data privacy attacks* and *model privacy attacks*. Concretely, data privacy attacks include membership inference attacks (MIA) (Shokri et al., 2017; Yeom et al., 2018; Chen et al., 2020a; Carlini et al., 2022; Xiang et al., 2024a;b) which aim to infer whether a given sample comes from training set or not, and data reconstruction attacks (DRA) (Fredrikson et al., 2015; Carlini et al., 2021; Geiping et al., 2020) which aim to extract exact training samples based on model outputs or gradients. Recent studies find that data privacy attacks can be enhanced through poisoning targeted models (Tramèr et al., 2022). On the other hand, model privacy attacks focus on extracting private information of models such as training hyperparameters (Wang & Gong, 2018), architectures (Ippolito et al., 2023; Oh et al., 2018; Finlayson et al., 2024; Carlini et al., 2024), model parameters (Rolnick & Kording, 2020; Jagielski et al., 2020), and functionalities (Tramèr et al., 2016; Orekondy et al., 2019; Truong et al., 2021). Our PEI attack also falls under the category of model privacy attacks. Unlike existing approaches, the PEI attack aims to infer a specific component of the architecture of downstream ML services, *i.e.*, the used pre-trained encoder.

**Privacy attacks against pre-trained encoders.** For data privacy attacks, many efforts have been made to design MIA against encoders. When the pre-training strategy is known, (Liu et al., 2021) successfully performs membership inference attacks (MIA) against targeted encoder based on membership scores calculated with the pre-training loss function and shadow training techniques. (He et al., 2022) extends MIA against semi-supervisedly learned encoders. (Zhu et al., 2024) studies a more challenging setting where the adversary has no knowledge about pre-training algorithms and proposes to calculate the embedding similarity between global and local features as the membership score for each sample. Besides, for model privacy attacks, studies mainly focus on stealing the powerful encoding functionality of encoders. By simply querying the targeted encoder and collecting return embeddings, (Liu et al., 2022b) succeeds in retraining an encoder that reproduces the functionality of the target with a reasonable query budget. (Sha et al., 2023) adopted a contrastive learning-based method to further improve the encoder stealing performance. However, these data/model privacy attacks can only threaten pre-trained encoders on the upstream side, while our PEI attack can further threaten downstream ML services.

**Encoder attacks against downstream models.** To increase the vulnerability of downstream models, adversaries usually try to attack upstream encoders by manipulating their pre-training stage with poisoned or backdoored training data. A series of works have studied poisoning/backdooring vision encoders when the downstream task is image classification. (Carlini & Terzis, 2022) and (Jia et al., 2022) showed that by injecting trigger patches to the training data, classifiers built upon the backdoored encoder can effectively misclassify inputs with backdoor triggers to a target label. (Carlini & Terzis, 2022) further found that only backdooring a significantly modest amount of training data (around $0.01\%$) is enough to perform strong attacks. Other advances include (Liu et al., 2022a), (Zhang et al., 2022) and (Tao et al., 2023). More recent works have started to exploit encoder backdoor attacks to increase the privacy vulnerability of downstream models built upon backdoored encoders, for example, making downstream training data more vulnerable to membership inference attacks (Wen et al., 2024) or data reconstruction attacks (Feng & Tramèr, 2024). However, these attacks usually require detailed information about the downstream tasks, such as the number of classes in downstream classification tasks or prior knowledge of potential targeted training data, which may limit their practicality in the real world. As a comparison, our PEI attacks can be performed in a downstream task-agnostic manner without modifying the pre-training stage of upstream encoders.

## C    OMITTED DETAILS OF SECTION 4

This section collects additional experimental details omitted from Section 4.

## C.1 ADDITIONAL EXPERIMENTAL SETUP

**Downstream datasets.** Three image datasets are used as downstream data: **CIFAR-10** (Krizhevsky, 2009) that consists of $50,000$ training images and $10,000$ test images from 10 classes; **SVHN** (Netzer et al., 2011) that consists of $73,257$ training images and $26,032$ test images from 10 classes; and **Food-101** (Bossard et al., 2014) that consists of $75,750$ training images and $25,250$ validation images (used as test data) from 101 classes.

**Pre-trained encoders.** Six vision encoders are adopted as candidates in the PEI attack, which are: **ResNet-34 (HF)** (He et al., 2016), **ResNet-50 (HF)**, **MobileNetV3** (Howard et al., 2019), **ResNet-34 (MS)**, **ResNet-50 (MS)**, and **CLIP ViT-L/14** (Radford et al., 2021). All these encoders can be downloaded from the Hugging Face Model Repository. Download links are collected and presented in Table 4.

Table 4: Download links of pre-trained vision encoders adopted in our experiments.

| Model Type | Name | Link |
|---|---|---|
| Vision Encoder | ResNet-34 (HF) | https://huggingface.co/timm/resnet34.a1_in1k |
| | ResNet-50 (HF) | https://huggingface.co/timm/resnet50.a1_in1k |
| | ResNet-34 (MS) | https://huggingface.co/microsoft/resnet-34 |
| | ResNet-50 (MS) | https://huggingface.co/microsoft/resnet-50 |
| | MobileNetV3 | https://huggingface.co/timm/mobilenetv3_large_100.ra_in1k |
| | CLIP ViT-L/14 | https://huggingface.co/openai/clip-vit-large-patch14 |

**Building downstream classifiers.** For each pair of encoder and dataset, we fix the encoder and train a downstream classifier based on embeddings of downstream training data obtained from the encoder. For each service, the classifier is an MLP consisting of three fully-connected layers, where the dimension of each hidden layer is $512$. We use Adam to train the classifier for $10,000$ iterations. The batch size is set as $512$. The learning rate is set as $0.001$ and decayed by $0.1$ every $4,000$ iterations.

## C.2 PEI OBJECTIVE IMAGES

For the PEI attack against vision encoders, we randomly sampled 10 images from the PASCAL VOC 2012 dataset (Everingham et al., 2012) as the objective image samples. They are presented in Figure 4.

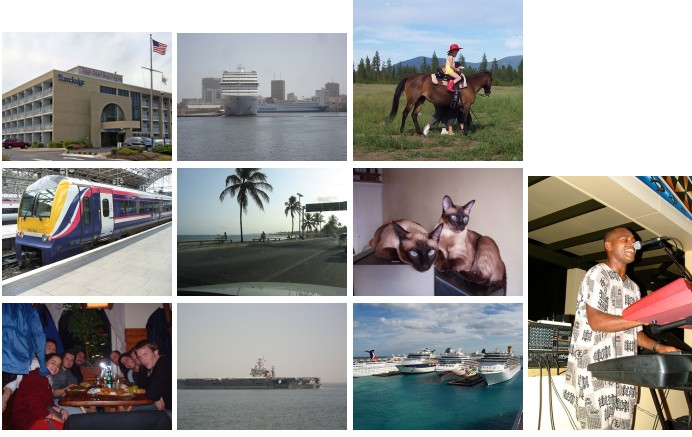

Figure 4: The 10 objective images used in the PEI attack against vision encoders.

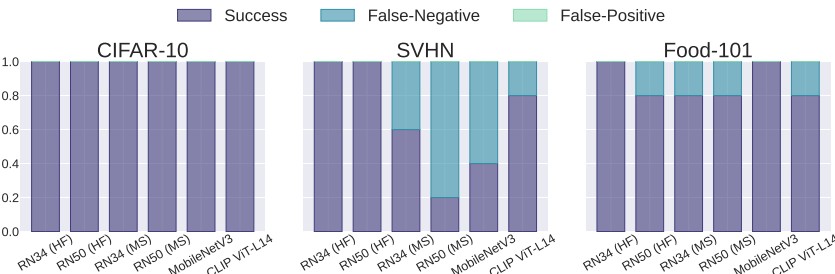

Figure 5: Rates of different PEI attack results on CIFAR-10, SVHN, and Food-101 datasets. Rates of attack success, false-negative, and false-positive are colored differently.

## C.3 DOWNSTREAM UTILITIES OF IMAGE CLASSIFICATION

We collect and present the classification accuracy of downstream image classification services built on different downstream data and pre-trained encoders. The results of downstream image classification services are reported in Table 5.

Table 5: Test accuracy (%) of downstream image classification services built upon different pre-trained encoders.

| Downstream Dataset | Upstream Pre-trained Encoder | | | | | |
|---|---|---|---|---|---|---|
| | RN34 (HF) | RN50 (HF) | RN34 (MS) | RN50 (MS) | MobileNetV3 | CLIP ViT-L/14 |
| CIFAR-10 | 91.24 | 91.88 | 90.44 | 91.05 | 91.34 | 98.11 |
| SVHN | 62.93 | 67.22 | 60.99 | 69.67 | 71.23 | 82.58 |
| Food-101 | 64.40 | 68.66 | 63.26 | 68.67 | 71.20 | 94.66 |

## C.4 ABLATION STUDIES OF SECTION 4.2

This section collects ablation studies omitted from Section 4.2.

**Effect of downstream classifier initialization.** The PEI attack in Table 1 is conducted against single downstream classifiers. Yet, it remains unknown how it would be affected by the random initialization of these classifiers. To investigate this, for each dataset-encoder pair, we retrain 5 downstream models, perform the PEI attack against all of them, and report rates of attack success, false-negative, and false-positive in Figure 5.

From the figure, we have two observations. Firstly, PEI attacks never made false-positive inferences in all analyzed cases. As explained in the previous section, this capability is useful for PEI adversaries to quickly find victims from a large number of targeted services. Secondly, in most of the cases (except two cases "RN34 (MS) + SVHN", and "MobileNetV3 + SVHN"), the attack achieved consistent results, *i.e.*, either attack success or false-positive, with a possibility of at least 80%, in 16 out of 18 analyzed cases. This suggests that the PEI attack is generally insensitive to the initialization of downstream classifiers in image classification services.

**Effect of downstream classifier architectures.** So far, the architecture of downstream classifiers used in image classification services is fixed to a 3-layer MLP with a width of 512. We now analyze how the architecture of this classifier would affect the PEI attack performance. The analysis is conducted on two cases: "RN34 (HF) + CIFAR-10" and "RN50 (MS) + CIFAR-10". For each case, we build classification services with the targeted encoder and four different downstream classifiers: MLP-1, MLP-2, MLP-3, and MLP-4. Here, "MLP-$x$" denotes an MLP architecture of $x$ layers, where dimensions of all hidden layers (if it has) are fixed to 512. The training of these downstream classifiers follows that described in Section C.1. After that, we attack each service with PEI attack samples originally used in Section 4.2. PEI $z$-scores are reported in Figure 6. From the figure, we find that the downstream classifier architecture has little effect on attacks against targeted services: the PEI $z$-scores of each candidate are almost the same across different downstream classifier architectures, and the PEI attack always revealed the correct hidden encoder.

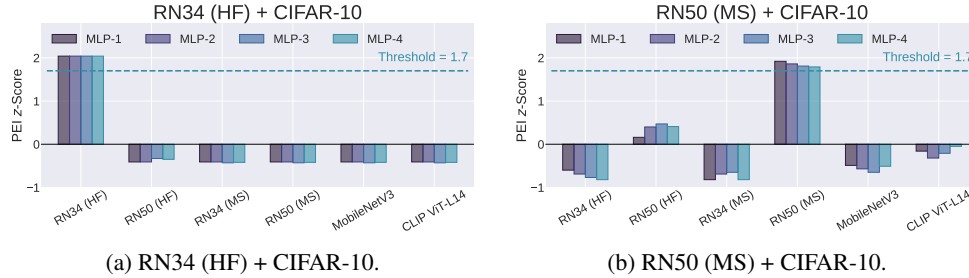

(a) RN34 (HF) + CIFAR-10.                    (b) RN50 (MS) + CIFAR-10.

Figure 6: PEI $z$-scores of candidates on classification services with downstream classifiers of 4 different architectures. $z$-scores of different services are in different colors.

**Effect of the zeroth-order gradient estimation on the PEI attack image synthesis.** According to Eq. (5), the gradient estimation accuracy depends on the random sampling number $S$. We thus vary $S$ in the set $\{25, 50, 75, 100, 120\}$ (in Section 4, $S = 100$) to see how the attack performance would change. This study is conducted on "RN34 (HF) + CIFAR-10" and "RN50 (MS) + CIFAR-10". PEI $z$-scores of different encoders under different $S$ are plotted in Figure 7. For the first case (Figure 7a), the sampling number $S$ has little effect on PEI $z$-scores, while the $z$-score of the correct hidden encoder always stays in high level (around 2.0). For the second case (Figure 7b), as $S$ increases, the PEI $z$-score of the correct encoder significantly increases, while those of other candidates remain at low levels. These imply that when the PEI attack is weak, it can benefit from a large sampling number $S$. However, since a large $S$ would increase the query budget, the adversary needs to carefully trade-off between the attack utility and efficiency.

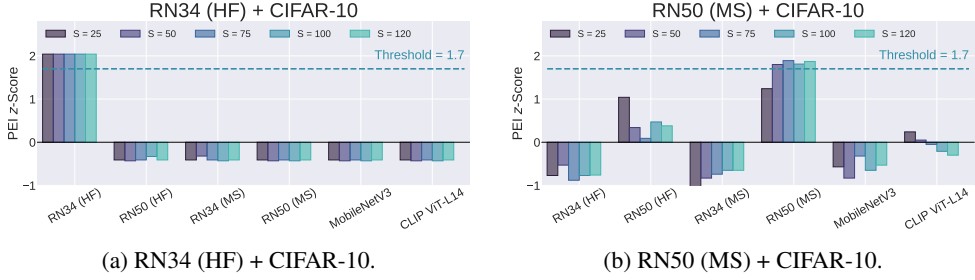

(a) RN34 (HF) + CIFAR-10.                    (b) RN50 (MS) + CIFAR-10.

Figure 7: PEI $z$-scores of candidates on image classification services with different gradient estimation sampling number $S$ (see Algorithm 2) varies in $\{25, 50, 75, 100, 125\}$.

### C.5    ADDITIONAL EXPERIMENTAL DETAILS OF SECTION 4.3

This section collects omitted experimental details of the PEI-assisted model stealing in Section 4.3.

**Building stolen models.** We train every stolen model via Adam for $10,000$ iterations, where the batch size is $64$ and the learning rate is initialized as $0.001$ and decayed by $0.1$ every $4,000$ iterations. The surrogate data is constructed from $50$ thousand images randomly selected from the validation set of ImageNet (Deng et al., 2009), which results in a query budget of $50$ thousand.

## D    OMITTED DETAILS OF SECTION 5

This section collects additional experimental details omitted from Section 5.

### D.1    PRE-TRAINED MODELS DOWNLOAD LINKS

The targeted downstream multimodal generative model is LLaVA-1.5-13B (Liu et al., 2023a;b), which is built upon a finetuned Vicuna-1.5-13B language model (Zheng et al., 2023) and the original CLIP ViT-L/14-336px vision encoder (Radford et al., 2021). We adopt seven vision encoders

to form the PEI candidates set, which are: **CLIP ViT-B/16 (Radford et al., 2021)**, **CLIP ViT-B/32**, **CLIP ViT-L/14**, **CLIP ViT-L/14-336px**, **OpenCLIP ViT-B/32 (Cherti et al., 2023)**, **OpenCLIP ViT-H/14**, and **OpenCLIP ViT-L/14**. All these models can be downloaded from the HUgging Face Model Repository. Download links are collected and presented in Table 6.

Table 6: Download links of pre-trained models adopted in Section 5.

| Model Type | Name | Link |
|---|---|---|
| Vision Encoder | CLIP ViT-B/16 | https://huggingface.co/openai/clip-vit-base-patch16 |
| | CLIP ViT-B/32 | https://huggingface.co/openai/clip-vit-base-patch32 |
| | CLIP ViT-L/14 | https://huggingface.co/openai/clip-vit-large-patch14 |
| | CLIP ViT-L/14-336px | https://huggingface.co/openai/clip-vit-large-patch14-336 |
| | OpenCLIP ViT-B/32 | https://huggingface.co/laion/CLIP-ViT-B-32-laion2B-s34B-b79K |
| | OpenCLIP ViT-H/14 | https://huggingface.co/laion/CLIP-ViT-H-14-laion2B-s32B-b79K |
| | OpenCLIP ViT-L/14 | https://huggingface.co/laion/CLIP-ViT-L-14-laion2B-s32B-b82K |
| LLaVA (Liu et al., 2023a;b) | LLaVA-1.5-13B | https://huggingface.co/llava-hf/llava-1.5-13b-hf |

### D.2 PEI SCORE CALCULATION

To calculate PEI scores following Eq. (2), one has to carefully design how to calculate the behavior similarity $\ell_{sim}(g(x_{i,j,k}^{(atk)}), g(x_j^{(obj)}))$ between PEI attack image $x_{i,j,k}^{(atk)}$ and $x_j^{(obj)}$ for the LLaVA model $g$. To this end, we propose calculating the behavior similarity by directly questioning LLaVA about the similarity between objective and attack images (in the range $[0, 1]$). Specifically, we ask LLaVA with the following prompt template:

```
USER: <image>\n<image>\nScore the similarity (in the range [0,1], higher
score means more similar) of the given two images\nASSISTANT:
```

where the two "<image>" in the prompt are placeholders for input images. We then construct the function $\ell_{ask}(\cdot, \cdot) : \mathcal{X} \times \mathcal{X} \to [0, 1]$ that maps the two (ordered) images to a similarity score based on LLaVA and the above prompt. The eventual similarity function $\ell_{sim}$ in Eq. (2) is defined as follows,

$$\ell_{sim}(g(x_{i,j,k}^{(atk)}), g(x_j^{(obj)})) := \frac{1}{2} \left( \ell_{ask}(x_{i,j,k}^{(atk)}, x_j^{(obj)}) + \ell_{ask}(x_j^{(obj)}, x_{i,j,k}^{(atk)}) \right).$$

### D.3 PEI-ASSISTED ADVERSARIAL EXAMPLE SYNTHESIS

To synthesize adversarial examples based on the PEI-revealed vision encoder (*i.e.*, CLIP ViT-L/14-336px), we propose to minimize their embedding difference with that of a targeted image containing preset harmful text content. Specifically, we aim to minimize the squared loss defined by embeddings of the adversarial image and targeted image. Since the hidden CLIP model is an open-source model, one can further leverage first-order gradient descent to optimize the objective squared loss. We use sign gradient to minimize the objective loss for $2,000$ iterations, in which the learning rate is fixed to $0.01$. Besides, to further improve the success rate of obtaining valid PEI-assisted adversarial images, for each targeted image, we will first synthesize 16 *candidate* adversarial images, and then choose a single image that enjoys the best attack performance among them as the eventual output.

## E PRICES OF ENCODER-AS-A-SERVICE IN THE WILD

To better estimate the cost of conducting PEI attacks in the wild, here we collect and list the prices of some common real-world EaaSs in Table 7. Combined with analyses in Sections 4 and 5, the estimated price of the PEI attack is no more than $100 per encoder.

Table 7: Pricing data of vision EaaSs in the wild.

| Type | Supplier | Model Name | Price | Link |
|---|---|---|---|---|
| Vision Encoder | Vertex AI (by Google) | multimodalembeddings | $0.0001 / Image Input | https://cloud.google.com/vertex-ai/generative-ai/pricing Accessed Date: 2025-05 |
| | Azure AI (by Microsoft) | Image Embeddings | $0.10 / 1,000 Transactions | https://azure.microsoft.com/en-us/pricing/details/cognitive-services/computer-vision/ Accessed Date: 2025-05 |
| | voyage-multimodal-3 (by Voyage AI) | Image Embeddings | $0.03 / 1,000 Images (200px × 200px) | https://docs.voyageai.com/docs/pricing#multimodal-embeddings Accessed Date: 2025-05 |

## F  POTENTIAL DEFENSES

Finally, in this section, we discuss several potential defenses against the PEI attack.

### F.1  TRANSFORMING PEI ATTACK SAMPLES

A possible direction of defending the PEI attack is to perform data transformation on PEI attack samples before feeding them to the protected downstream service to sanitize their harmfulness. While many existing filtering-based methods (Liao et al., 2018; Nie et al., 2022) could be used to remove malicious information in PEI attack samples, they may also reduce model performance on clean data. Meanwhile, a series of works also focus on making adversarial examples robust to data transformation (Athalye et al., 2018; Fu et al., 2022; Chen et al., 2023). As a preliminary investigation, here we analyze a simple *plug-and-play* and *downstream task-agnostic* sample sanitizing methods for vision encoders.

**A plug-and-play preprocessing defense for vision encoders.** As PEI attack samples can somewhat be seen as adversarial examples (Goodfellow et al., 2015), one may naturally seek to adopt defenses originally proposed for adversarial attacks to tackle the new PEI attack. As a preliminary investigation, here we adopt a task-agnostic and plug-and-play method named *JPEG defense* (Guo et al., 2017) to protect targeted services from PEI attack. Concretely, for each query image, the service supplier will first process it with the canonical JPEG compressing algorithm before feeding it into the real downstream service $g$. As JPEG compression is found to be effective in destroying harmful information in adversarial images (Guo et al., 2017; Dziugaite et al., 2016), it might also be able to mitigate hazards of PEI attack images. We conduct case studies on image classification services "RN34 (HF) + CIFAR-10" and "RN50 (MS) + CIFAR-10".

The results of JPEG defense against the vanilla PEI attack are presented in Figure 8, in which we plot $z$-scores of different candidates with and without the JPEG defense (denoted as "Origin" and "JPEG-Def" respectively). We find that the JPEG defense effectively defeats the PEI attack: in all cases, under the JPEG defense, the PEI $z$-scores of the correct hidden encoder are suppressed to low levels and no candidate goes beyond the preset threshold 1.7. But we also discovered a simple method to bypass the defense for PEI attack: we rescale each PEI attack image from $64 \times 64$ to $512 \times 512$ before sending them to the target service. Results are also presented in Figure 8 (denote as "JPEG-Def vs. Resize-512"), which show that this rescaling approach enables the PEI attack to succeed again in all analyzed cases. We deduce this is because resizing an attack image to a larger scale can prevent local features that contain malicious information from being compressed by JPEG defense.

### F.2  DETECTING PEI ATTACK SAMPLES

Another potential direction is to leverage Out-of-distribution (OOD) detection to identify and drop malicious queries toward protected ML services. For image data, a series of detection methods (Tran et al., 2018; Pang et al., 2018; Dong et al., 2021; Li & Li, 2017; Roth et al., 2019) against adversarial images or backdoor images could be applied to defend against the PEI attack. However, PEI attack samples can also be made *stealth* to bypass potential detection defenses. For example, for image data, one may be able to exploit detection bypassing methods originally designed for adversarial

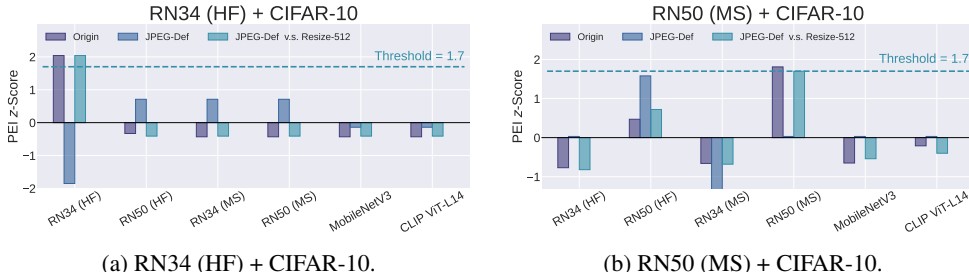

(a) RN34 (HF) + CIFAR-10.          (b) RN50 (MS) + CIFAR-10.

Figure 8: PEI $z$-scores of candidates on image classification services. (1) "**Origin**": original PEI attack results. (2) "**JPEG-Def**": results of PEI attack vs. JPEG defense. (3) "**JPEG-Def vs. Resize-512**": results of PEI attack strengthened by resizing (to $512 \times 512$) vs. JPEG defense.

images (Bryniarski et al., 2022; Carlini & Wagner, 2017; Hendrycks et al., 2021) to PEI attack images.

### F.3  ARCHITECTURE RESISTANCE

More reliable defenses may be re-designing the pipeline of building downstream ML services to make them natively robust to the PEI attack. For example, as the current PEI attack assumes that the targeted downstream service is built upon a single encoder, one may leverage multiple upstream encoders to build a single service to bypass the attack. The idea of using multiple pre-trained encoders has already been adopted to improve the downstream model performance (*e.g.*, SDXL (Podell et al., 2023)). Our results suggest that apart from improving performance, there is a great need to use multiple encoders for the sake of downstream model security.

