# OpenReview forum: "Pre-trained Encoder Inference: Revealing Upstream Encoders In Downstream Machine Learning Services"
_ICLR.cc/2026/Conference — ICLR 2026 Conference Withdrawn Submission_

### Official Review · Reviewer_xjhU · 2025-10-15

**Soundness:** 2
**Presentation:** 3
**Contribution:** 3
**Rating:** 2
**Confidence:** 4

**Summary:**

This paper introduces the Pre-trained Encoder Inference (PEI) attack, a new class of encoder privacy attacks that aims to reveal which pre-trained encoder is secretly used inside a downstream ML service. The authors evaluate PEI on image classification models and LLaVA, and further show how knowing the encoder can facilitate downstream model stealing and adversarial attack.

**Strengths:**

1. The paper highlights that encoder privacy can still be compromised in downstream services
2. The proposed method is adapted to various image encoders as demonstrated in experiments

**Weaknesses:**

1. The authors claimed that their method is downstream-task agnostic. However, the proposed algorithm critically relies on continuous input spaces (e.g., pixel-level perturbations) and thus seems tailored to vision encoders. For text encoders with discrete token inputs, the optimization strategy cannot be directly applied, raising doubts about the claimed generality.

2. I noticed that the datasets tested in the article are all low-resolution (e.g., CIFAR-10, SVHN), which may be one of the reasons for the query overhead ($100 per encoder). It is necessary to discuss the effect of image resolution on the scale of S in Equation 5.

3. A larger candidate encoder set naturally increases the chance of covering the hidden encoder, but it also significantly raises the query cost for synthesizing and testing PEI samples. The paper does not clearly address how to balance this trade-off.

4. The paper’s Table 3 shows that knowing the encoder can improve adversarial transfer. However, this result depends on the revealed encoder (CLIP) being available for white-box use. It is unclear whether the same advantage holds when the encoder itself is a black-box (no open weights and only limited API access). Intuitively, if both the encoder and the downstream model are black boxes, a direct black-box attack on the downstream model may be more query-efficient and reliable than the three-step pipeline (PEI → encoder-level black-box/stolen surrogate → transferred white-box attack). The paper should temper claims about general gains or provide targeted experiments under fully black-box conditions.

**Questions:**

None

---

### Official Review · Reviewer_CEn6 · 2025-10-27

**Soundness:** 3
**Presentation:** 3
**Contribution:** 2
**Rating:** 2
**Confidence:** 4

**Summary:**

This paper proposes Pre-trained Encoder Inference (PEI): a method for inferring the hidden pre-trained encoder underlying a downstream service when only API access is available—specifically in an Encoder-As-A-Service setting where the downstream model is invisible—and thereby significantly enhancing the effectiveness of model stealing and adversarial attacks. The paper systematically validates the feasibility of PEI on image classification and multimodal generation (LLaVA) and provides preliminary analyses of its cost, robustness, and potential defenses.

PEI consists of two main phases. In the first phase, given a candidate encoder $f_i$ and a current input sample, an adversarial PEI sample is synthesized such that, under the candidate encoder $f_i$, the embedding of the adversarial sample is as close as possible to that of the original sample. In the second phase, this adversarial sample is fed into the downstream service, and similarity scores are computed. Finally, a one-sided z-test is employed to identify the correct encoder.

**Strengths:**

1. The experiments are comprehensive and rigorous, with a low false positive rate. The method successfully identifies the correct encoder in 16 out of 18 image classification services. Importantly, it does not produce false positives in the two failure cases, indicating statistical robustness according to the paper’s results.

2. The paper also explores downstream applications of the inferred encoder, such as enhancing model stealing and adversarial attacks, forming a complete and coherent attack pipeline.

3. The paper is well-structured and clearly presented.

**Weaknesses:**

1. It is unclear why the authors construct adversarial examples in this way. Why do embedding-similar / visually-different samples induce distinguishable behaviors on the downstream task? The explanation provided in the paper is not clear enough and needs further elaboration.

2. The method assumes that the attacker possesses a candidate set E containing the true hidden encoder, or at least models sufficiently similar (e.g., from the same model family). However, it remains unclear whether the approach can reliably infer the encoder when the service employs a proprietary or privately fine-tuned encoder. The paper currently lacks a systematic evaluation and discussion of this point. This is a key limitation, as many service providers fine-tune models according to their specific business needs.

3. The paper mentions that the candidate set can be small (since, in reality, EaaS is provided by only a few companies), but this does not cover the actual situation, such as internally customized models, different fine-tuned versions, or variants of the same architecture trained on different datasets. In practice, the number of possible encoders in the real world is much larger.

4. The paper lacks a systematic comparison with current mainstream encoder or model-stealing attack methods, such as: Can’t Steal? Cont-Steal! Contrastive Stealing Attacks Against Image Encoders (CVPR '23) and MAZE: Data-Free Model Stealing Attack Using Zeroth-Order Gradient Estimation (CVPR '20)

5. Moreover, the paper claims that knowing the encoder helps subsequent attacks, but in fact, numerous studies have shown that attackers can successfully perform functional cloning, model stealing, or other attacks using only a publicly available black-box API (input => prediction), even without knowledge of the encoder’s architecture or identity. The paper should include comparisons against relevant baselines, such as: Stealing Hyperparameters in Machine Learning (S&P '18) and Simulating Unknown Target Models for Query-Efficient Black-Box Attacks (CVPR '21).

6. Practical deployment against real-world APIs may face challenges. Under realistic budget constraints, testing against an actual service would be valuable, as real APIs often impose quota limits, rate limiting, and A/B testing. Since the attack requires a large number of API queries, the associated cost and feasibility in practice could be substantial.

7. Minor Issues:
   - Typos: “pervious” => “previous”.
   - Inconsistent notations (e.g., “RN-50” vs. “RN50”).
   - Inconsistent verb tenses throughout the paper. Some sections use the present tense while others use the past tense.

**Questions:**

See weaknesses

---

### Official Review · Reviewer_Uw1M · 2025-10-29

**Soundness:** 3
**Presentation:** 2
**Contribution:** 2
**Rating:** 4
**Confidence:** 4

**Summary:**

The paper introduces a novel attack called Pre-trained Encoder Inference (PEI), which targets downstream machine learning (ML) services that use hidden pre-trained encoders. Unlike existing attacks that compromise encoders during pre-training, PEI operates in a black-box setting. The adversary only has API access to both the downstream service and a few candidate encoders from model repositories. The core idea is to synthesize PEI attack samples whose embeddings match a reference embedding only under a specific encoder. By evaluating how a downstream service reacts to these synthesized samples, the adversary can infer which pre-trained encoder (if any) is secretly used. Extensive experimental results demonstrate that the proposed attacks can effectively infer the inner information of the hidden encoder.

**Strengths:**

- Novel attacks
- Well-written paper
- Comprehensive evaluastion

**Weaknesses:**

- Limited Scope of Modalities
- Assumption on Candidate Set Availability
- Unpractical motivation

**Questions:**

This paper introduces the new kinds of model infer attacks against the image encoders. The authors generate several special images which can reveal different characteristics of different encoders. These images can be used to infer which encoder is used in the hidden service. The paper is well-written and the epxiermental results are convicing. However, I do have some concerns regarding the motivation and the practical value of the attack.

- The proposed attacks rely on the presence of image-based services for classification or generation. To my knowledge, however, such services are currently rare or not sufficiently widespread. The authors should therefore strengthen their argument for the practicality of the attack by explaining realistic deployment scenarios, quantifying the prevalence of vulnerable services, or discussing how the attack would apply to less common or private deployment models.

- The paper focuses solely on vision encoders and one multimodal model (LLaVA). There is no demonstration on language, audio, or embedding-as-a-service APIs, where the attack might behave differently.

- The success of PEI heavily depends on having access to the true encoder or close variants in the candidate pool. Real-world systems may use custom fine-tuned or privately modified encoders, reducing practical applicability. The authors are supposed to explain how the proposed attacks work when this situation happened.

---

### Official Review · Reviewer_SXd6 · 2025-10-30

**Soundness:** 3
**Presentation:** 3
**Contribution:** 3
**Rating:** 6
**Confidence:** 3

**Summary:**

This paper introduces PEI (Pre‑trained Encoder Inference), a reverse‑lookup attack that infers the pre‑trained encoder running inside a downstream ML service from black‑box API interactions. The attacker queries the service, assembles a candidate encoder pool, and compares responses to narrow down the likely encoder. Unlike prior attacks that assume access to upstream models, PEI aligns more closely with production settings—making it operationally messier but practically more informative.

**Strengths:**

1.The paper proposes a new attack,PEI, that pivots attention from classic upstream pre-trained encoders to the ones actually tucked inside production, downstream ML services. This reframed threat model tracks real deployment practice, fills an obvious hole in prior work, and surfaces a vulnerability that has been hiding in plain sight.

2.The authors validate PEI across image classification and a multimodal generation service (LLaVA). On 18 image-classification targets, it correctly uncovered the internal encoder for 16 services with very high precision.

3.The z-score thresholding technique was ingeniously introduced, achieving a low false positive rate: when a correct encoder exists, it can be accurately identified, and when no encoder is present, false positives are rare (only 1 out of 6 cases on CIFAR-10). This allows the resulting probabilities, under a reasonable Gaussian assumption, to distinguish between recognized and unrecognized cases.

**Weaknesses:**

1、A key fragility is PEI’s sensitivity to input preprocessing and the narrow assumptions behind the proposed bypass. Although the paper notes that PEI fails under common transforms (e.g., JPEG), the suggested defense—enlarge the image before querying—implicitly assumes the server will apply JPEG to the enlarged image before resizing to the model input. If the server instead resizes first then applies JPEG, or applies an unconventional, hard‑to‑predict processing pipeline, the bypass fails. The attack’s success therefore appears to depend heavily on the exact ordering of these operations, an ordering the attacker cannot reliably control or verify in practice. The authors should evaluate robustness across different realistic preprocess orderings and discuss how an attacker could (if at all) detect or enforce the server’s processing pipeline.

2、Experiments focus primarily on ResNet / CLIP vision encoders. Portability to substantially different blueprints (proprietary or non‑standard architectures) and to other modalities (text, audio, etc.) remains unestablished. The manuscript needs additional evidence that PEI’s similarity signals continue to separate candidates when architectures—and thus inductive biases—diverge. At minimum, the authors should (a) test a few non‑standard or fine‑tuned variants, and/or (b) discuss limitations and expected failure modes for other modalities.

3、PEI implicitly assumes the attacker’s preprocessing (TransformA) yields inputs whose embeddings—under a candidate encoder—are comparable to embeddings produced by the service’s hidden preprocessing (TransformB). The paper does not adequately stress‑test $A \neq B$ scenarios (color space, normalization, crop/resize policy, tokenization, etc.), which in practice can serve as a strong defense knob. Misalignment between $A$ and $B$ will push embeddings apart and degrade inference reliability. The authors should quantify PEI’s sensitivity to typical preprocessing mismatches and propose mitigation strategies or calibration procedures.

4、The method assumes in its design that the hidden encoder f* is within the candidate set E; if f* is not included, the strategy can only conclude with an "empty set." However, in practice, it is more common for "family neighbors" to cause false matches with high scores that are not the true f* (for example, different checkpoints or fine-tuned versions of CLIP). Additionally, during the synthesis phase, M2 adversarial samples need to be generated for each candidate and each target sample, so the overall query and computational cost is proportional to N×M1×M2. This can be a significant obstacle for services with rate limits, added noise, or randomized responses. The choice of similarity function ℓ_sim and noise in service outputs also directly impact statistical power. It is recommended to add optimization techniques to minimize the number of queries required for candidate models as much as possible.

5、All candidate encoders must synthesize adversarial samples from the same initialization for each target; otherwise the PEL score may conflate model performance with independent random variability in initialization. The paper should explicitly enforce and report a consistent initialization procedure (and seed control) and evaluate the sensitivity of PEL to initialization noise.

**Questions:**

See Weakness

---

### Note · Authors · 2025-11-13

I have read and agree with the venue's withdrawal policy on behalf of myself and my co-authors.